# Recent Progress on Micro-Fabricated Alkali Metal Vapor Cells

**DOI:** 10.3390/bios12030165

**Published:** 2022-03-06

**Authors:** Xuelei Wang, Mao Ye, Fei Lu, Yunkai Mao, Hao Tian, Jianli Li

**Affiliations:** 1School of Instrumentation and Optoelectronic Engineering, Beihang University, Beijing 100191, China; wangxuelei@buaa.edu.cn (X.W.); lufei@buaa.edu.cn (F.L.); maoyunkai6@buaa.edu.cn (Y.M.); hmysth@163.com (H.T.); lijianli@buaa.edu.cn (J.L.); 2Beihang Hangzhou Innovation Institute Yuhang, Hangzhou 310023, China

**Keywords:** micro-fabrication, alkali vapor cell, atomic sensors, core component

## Abstract

Alkali vapor cells are the core components of atomic sensing instruments such as atomic gyroscopes, atomic magnetometers, atomic clocks, etc. Emerging integrated atomic sensing devices require high-performance miniaturized alkali vapor cells, especially micro-fabricated vapor cells. In this review, bonding methods for vapor cells of this kind are summarized in detail, including anodic bonding, sacrificial micro-channel bonding, and metal thermocompression bonding. Compared with traditional through-lighting schemes, researchers have developed novel methods for micro-fabricated vapor cells under both single- and double-beam schemes. In addition, emerging packaging methods for alkali metals in micro-fabricated vapor cells can be categorized as physical or chemical approaches. Physical methods include liquid transfer and wax pack filling. Chemical methods include the reaction of barium azide with rubidium chloride, ultraviolet light decomposition (of rubidium azide), and the high-temperature electrolysis of rubidium-rich glass. Finally, the application trend of micro-fabricated alkali vapor cells in the field of micro-scale gyroscopes, micro-scale atomic clocks, and especially micro-scale biomagnetometers is reviewed. Currently, the sensing industry has become a major driving force for the miniaturization of atomic sensing devices, and in the near future, the micro-fabricated alkali vapor cell technology of atomic sensing devices may experience extensive developments.

## 1. Introduction

Alkali vapor cells are the core components of atomic sensors, including atomic gyroscopes, atomic clocks, and atomic magnetometers [1,2,3,4,5]. Traditional alkali vapor cells are mostly composed of six pieces of glass joined through bonding techniques, which fabricate a square vapor cell, or fabricate a spherical vapor cell through the traditional glass-blowing technique. Glass blowing is advantageous because this technology is well developed with a significantly lower cost compared to bonding technology. Notably, certain problems exist in the manufacture of alkali vapor cells with flat and clean surfaces to meet the requirements for a high transmission rate, for instance, square alkali vapor cells. These processes, in turn, vary the properties of each piece of glass, in terms of thickness, parallelism, and flatness, which induces unexpected inconsistency. Furthermore, spherical alkali vapor cells will have a scattering effect on the light beams. More importantly, traditional alkali vapor cells fabricated through the blowing method cannot be further miniaturized; this method limits the further miniaturization of atomic sensors. Thus, micro-fabricated alkali vapor cells have been developed and become an important component for miniaturized atomic sensors.

The requirements for miniaturization, integration, and high precision in magnetometers have become the major driving force for the development of miniaturized atomic vapor cells, which are the core component and also known as the heart of atomic sensors. The size of this vapor cell directly limits the final volume of the whole device. With the application of micro-/-nano-fabrication technology [6,7,8], the volume of vapor cells can be reduced to the cubic centimeter or even cubic millimeter level, which facilitates the integration of core components.

The application of micro-fabricated vapor cells is mainly in micro-fabricated atomic gyroscopes [9,10,11,12,13], micro-fabricated atomic clocks [14,15,16,17,18], and micro-fabricated atomic magnetometers [19,20,21,22]. The most significant development of micro-fabricated alkali vapor cell technology lies in the field of micro-miniature atomic magnetometers owing to the urgent demand in the fields of biomagnetic measurements [23,24,25,26,27,28], magnetoencephalography (MEG) and magnetocardiography (MCG) [29,30].

## 2. Recent Progress on Micro-Fabricated Alkali Vapor Cells

### 2.1. Bonding Methods

Depending on different fabrication methods, the anodic bonding of alkali metal vapor cells can be performed through various processes. Several major bonding methods exist: direct glass–silicon bonding (Figure 1), bonding by glass-frit reflow, and metal thermocompression bonding by plating metal films of different materials [31].

#### 2.1.1. Glass–Silicon–Glass Anodic Bonding

In 2011, Lv et al. of Wuhan University of Technology proposed a method for the glass–silicon–glass anodic bonding of two-electrode atomic vapor cells based on vacuum encapsulation [32]. A new process for fabricating atomic vapor cells utilizing laser perforation and surface analysis was proposed. A tensile strength test showed that the tensile strength of the specimens could reach 9–10 MPa. Spectral analysis showed that the alkali metal was well packaged. This article focuses on the application of coherent population trapping (CPT) atomic clocks and analyzes the importance of the long-term polishing of silicon under laser drilling.

In 2012, Pétremand et al. of the Institute of Microengineering at the Ecole Polytechnique Fédérale de Lausanne (EPFL) proposed a new technique for fabricating alkali metal vapor cells larger than micro-mechanical vapor cells and smaller than glass-blown vapor cells [33]; this technique was based on the anodic bonding of silicon and relatively thick glass wafers, thus covering the gap between 2 mm micro-fabrication techniques and 6–10 mm or larger classical glass-blowing techniques. The method proposed in this article can solve the problem of silicon thickness in the bonding process, and the manufactured alkali metal vapor cells can be utilized for atomic clocks and magnetometers.

In 2019, Zhang et al. of North University, China, described a deep reaction based on ion etching and vacuum anodic bonding techniques for the micro-machining of wafer-level vapor cells for MEMS [34]. With the gradual increase in voltage, the anodic bonding process at 200 V has a critical impact on alkali vapor cell packaging. In addition, the Si–glass bonding surface was experimentally investigated by scanning electron microscopy, which showed no visible cracks or defects on the bonding surface. The leakage rate was measured by utilizing a helium leak detector. The results showed that the vapor cells with different optical cavity lengths comply with the MIL-STD-883E standard (5 × 10^−8^ mbar·L/s). This article mainly describes the process and detection method for the synthesis of alkali metal vapor cell by bonding silicon and glass anodes. 

In 2020, Boudot et al. at the University of Colorado, USA, fabricated silicon glass micro-machined ultra-vacuum (UHV) vapor cells with silicon-etched cavities with non-evaporable getter pumps (NEG) and aluminosilicate glass (ASG) windows [35]. 

In 2021, the design, fabrication, and evaluation of sealed rubidium vapor cells by utilizing two-step bonding were conducted by Hitoshi Nishino et al. [36], Tohoku University. The proposed method separates the optical path filled with alkali atoms and the alkali dispenser chamber in a wafer-level process. This article mainly describes the application of the two-step anodic bonding of alkali metal vapor cells in atomic clocks.

The advantage of glass–silicon anodic bonding is that the technology is more developed; however, the disadvantage is that the bonding environment involves a high temperature, high voltage, and certain vacuum level, which requires special equipment such as wafer bonding tools.

#### 2.1.2. Sacrificial Micro-Channel Bonding

In 2011 [37] and 2013 [38], Tsujimoto et al. of the Department of Microengineering, Kyoto University, Japan, proposed a novel and simple encapsulation approach in which a sacrificial micro-channel at the bonding interface was utilized as a gas pathway for evacuation and filling, followed by reflow sealing with the glass material (Figure 2). The sealing enhances the gas pressure controllability of the micro-machined alkali vapor cell for chip-scale atomic magnetometers (CSAMs). A 10 mm^3^ potassium vapor cell filled with 0.1 MPa helium buffer gas was fabricated and sealed with a helium leak rate of less than 3.1 × 10^−14^ Pa·m^3^/s. This article proposes a new bonding method that can be utilized to make an alkali metal vapor cell that meets the requirements of magnetometers but has higher requirements for vacuum and bonding pressure.

The advantage of the sacrificial micro-channel glass-frit bonding method is that the exhaust gas generated during the introduction of an alkali metal can be eliminated in a timely manner; on the other hand, the working gases, such as the buffer gas required in the micro-fabricated alkali metal vapor cell, can be filled based on design, which is easy to control. However, the disadvantage of this is that the micro-channels need to be heated to the softening point after the introduction of the gas, and the micro-channels are sacrificed due to pressure, which completes the bonding operation. Due to the high temperatures required to reach the glass-frit softening point, the bonding process places high demands on the performance of the experimental equipment.

#### 2.1.3. Metal Film Thermocompression Bonding

In 2012 and 2013, Straessle et al., Ecole Polytechnique de Lausanne, Switzerland, [39] and Straessle et al., Neuchatel, Switzerland, utilized indium thin film to bond alkali metal vapor cells (Figure 3), respectively [40]. The requirement of temperature is as low as ≤140 °C to ensure a sufficient bonding strength, which makes it possible to carry out alkali metal vapor cell coating in nuclear magnetic resonance (NMR) gyroscopes.

In 2019, Wang et al. at the Center for Ultra-Precision Optoelectronic Instruments, Harbin Institute of Technology, proposed and designed an Au-In-TLP bonding technique (Figure 4) with an asymmetric metal structure for encapsulating wafer-level vapor cells at a low processing temperature of 200 °C [41]. The bonded vapor cell can withstand post-processing temperatures of up to 490 °C, with the shear strength of the bonded surface reaching 31.68 MPa. The upper limit of the leakage rate is 5 × 10^−10^ mbar·L·s^−1^ for packaged cells, which shows that the technology is suitable for low-temperature, hermetically packaged, and reliable wafer-level packaging for alkali metal micro-chamber fabrication. This article proposes a new bonding and packaging method, which can reduce the process temperature to below 200 °C and reduce the difficulty of bonding.

Micro-fabricated vapor cells are usually sealed via the anodic bonding of borosilicate to silicon, which is proved to be a reliable method but also suffers from specific limitations, including cavity inherent oxygen contamination or the limitations pertaining to several material combinations. As an alternative, in 2019, the successful fabrication of cells at the wafer level via Cu-Cu thermocompression bonding (Figure 5) was presented [42]. This alternative technique overcomes the limitations of anodic bonding, while allowing the application of new materials.

The advantages of metal thin-film bonding include overcoming the aforementioned requirements for anodic bonding at high temperatures, including sacrificial micro-channel bonding. In addition, it counters the drawback that common anodic bonding can only be performed by utilizing specific materials, enabling the bonding and operation of different materials under low-temperature conditions. The disadvantage is that the technology is not very developed at present, and further research is required.

### 2.2. Design of Light-Passing Scheme for Alkali Vapor Cells

The conventional planar structure of micro-fabricated alkali metal vapor cells only has two planes for light transmission, which only allows for the propagation of a single beam. For atomic sensors that require a double-beam pass-through scheme, two mutually perpendicular pumping and detection beams are required. The sandwich-structured glass–silicon–glass-bonded alkali metal vapor cell only has two light transmission planes, which does not meet the demand of two vertical beams projected onto the alkali metal vapor cell; thus, other alkali metal vapor cell structures need to be designed. There are two main methods to solve the demand for a dual beam: one is to utilize a micro-fabricated spherical vapor cell, which can enable the selection of a path with an improved light transmission rate to achieve the dual-beam solution, and the other is to utilize micro-fabrication technology or partial bonding to form a reflective structure inside the flat alkali metal vapor cell, thus enabling the beam to change direction inside the alkali metal vapor cell and meet the demand for two vertical light beams. Cylindrical glass vapor cells are mostly utilized for single-item technology verification or principle verification, and the application of miniaturization and integration requires that cylindrical vapor cells are not commonly utilized in micro-fabricated vapor cells [43], and will not be elaborated on here.

#### 2.2.1. Spherical Alkali Vapor Cells through Wafer Blowing

The processes of manufacturing spherical alkali metal vapor cells via the wafer-blowing method are as follows: First, a cubic or cylindrical cavity is etched in silicon, and then a thin glass sheet is anodically bonded with a silicon sheet engraved with the cavity. Next, the bonded wafers are placed in a furnace, with the temperature set at the glass softening point. As the viscosity of the glass decreases at this temperature, the heated residual gas in the cavity expands, such that the glass is blown into a three-dimensional spherical shape.

In 2007, Eklund et al. from the University of California, Irvine, invented the wafer-blowing method to manufacture glass shell arrays at the wafer level [44]. An analytical model of the glass shell shape was also established (Figure 6), and the sphericity of the spherical vapor cell and the uniformity of the glass shell thickness were analyzed and optimized. In 2008, Eklund et al. utilized 100 μm-thick borosilicate glass wafers at 850 °C to manufacture a spherical glass vapor cell. The wafer-blowing process [45] includes deep reactive ion etching, bonding, blowing, and annealing.

In 2013, Chen et al. of Southeast University fabricated a double-chambered spherical alkali metal vapor cell utilizing the wafer-blowing process [46], with the smaller chamber serving as the location for the chemical reaction to fabricate alkali metal and the larger chamber serving as the sensor’s sensing area, connected by micro-channels. Although there has been a lot of research on glass blowing vapor cells, square-structured vapor cells are mostly utilized in practical applications. 

In 2015, Shang et al. of Southeast University fabricated micro-spherical rubidium vapor cells (Figure 7) utilizing the wafer-blowing process [47]. The integration of a micro-spherical rubidium vapor cells with a three-axis atomic magnetometer was successfully demonstrated. The manufactured vapor cell is mainly utilized in atomic magnetometers.

In 2017, Ji et al. of Southeast University utilized a micro-spherical alkali metal vapor cell instead of a conventional glass cell (Figure 8) [48]. A dual-chamber vapor cell was designed to separate the alkali metal delivery chamber from the light detection chamber, and a chip-scale atomic magnetometer with tenfold higher sensitivity was achieved experimentally. The improvement in the spherical cells over flat cells is due to the fact that the optical path in spherical cells is much longer and there is a greater interaction between the light and the gas in the spherical chamber. It was proposed that the micro-spherical vapor cell has better relaxation resistance, especially at a low buffer gas pressure. This manufacturing method affects the range of buffer gas pressure values.

In 2018, Ji et al. of Southeast University designed a micro-machined spherical rubidium vapor cell with the same volume as a conventional micro-machined planar rubidium vapor cell [49]. This was interconnected with micro-channels to obtain equal internal buffer gas pressures. In the chip-scale spin-exchange relaxation-free (SERF) atomic magnetometer, the transverse polarization lifetimes of the spherical cells were 2.46, 1.44, and 1.05 times those of the planar cells at different buffer gas pressures of 0.35, 0.73, and 2.0 amg, respectively. Furthermore, the sensitivity of the chip-scale SERF atomic magnetometer of micro-spherical cells was 60 fT/Hz^1/2^, from 5 Hz to 15 Hz, while the sensitivity of planar-based atomic magnetometers was 400 fT/Hz^1/2^, from 5 Hz to 15 Hz. The manufacturing method in this article is suitable for magnetometer applications.

In 2019, Noor et al. utilized a micro-glass-blowing process to fabricate micro-sphere-type alkali metal vapor cells [50,51] and discussed a variety of design considerations, including chamber geometry, optical properties, materials, and surface coatings. The wafer-level coating process utilizing Al_2_O_3_ was found to increase the ^131^Xe relaxation time (T_2_) by a factor of 4, and T_2_ was improved by a factor of 3 after converting borosilicate glass (Pyrex) to aluminosilicate glass (ASG). The improvement in T_2_ is expected to reduce the ARW of NMR gyroscopes using Al_2_O_3_-coated vapor cells and ASG vapor cells by factors of 4 and 3, respectively. This method meets the needs of the dual-beam-scheme gyroscope.

Micro-fabricated spherical vapor cells meet the dual-beam requirement, but the spherical vapor cells have the disadvantages of difficulty in ensuring uniformity and overcoming the divergence of the beam.

#### 2.2.2. Anodic Bonding Square Alkali Vapor Cell

In 2003, Kitching et al. of NIST, USA, fabricated atomic vapor cells utilizing a combination of silicon wafer etching and anodic bonding methods in the micro-fabrication process [52]. The size of the vapor cell plane was 9 × 9 mm^2^, the radius of the hole was 750 μm, and the depth was 375 μm. In 2008, California Microsystems Laboratory [30], USA, prepared a reflector with multilayer non-metallic layers on the sidewalls of an inclined Rb vapor cell utilizing a multilayer amorphous silicon and silicon dioxide processing technique. The laser light was passed through the atomic vapor cell again and returned to the plane where the light source was located with a smaller optical power loss, and the optical return efficiency was improved by a factor of eight compared to the silicon reflector alone.

In 2014, NIST Ricardo et al. introduced the micro-machining vapor cell (Figure 9) [53]. The division into multiple functional chambers allows the simultaneous action of two optical paths: detection light and pumping light. This scheme has some problems with the practical application of the gyroscope. 

In 2015, the FEMTO-ST Institute and Ravinder Chutan reported the primitive structure of a micro-machined alkali vapor cell designed for miniature atomic clocks (Figure 10) [54]. The cell combines a diffraction grating with anisotropically etched single-crystal silicon sidewalls to pass a normally incident beam through a cavity oriented along the plane of the substrate. The grating is specifically designed to diffract circularly polarized light at one level, with the latter having a diffraction angle matching the (111) sidewall orientation. In this manner, the length of the cavity where the light interacts with the alkali atoms can be extended. This kind of scheme provides a very good basis and has good maneuverability for the design of our double-beam gyroscope. 

In 2019, Hishino et al. of Tohoku University, Sendai, Japan, proposed a rubidium vapor cell fabricated via the local anodic bonding of small 45° mirrors (Figure 11) [55]. A 45° reflector with a Bragg mirror was fabricated through a scribing process. A horizontal chamber vapor cell with 45° mirrors at both ends was fabricated through an anodic bonding process. The local bonding of the mirror seems to be difficult to operate. 

The planar structure reflected beam design scheme can realize the demand of double-beam vertical distribution inside the alkali metal vapor cell, but it has the disadvantages of difficult processing and technology that is not fully developed.

### 2.3. Alkali Metal Packaging Methods in Micro-Fabricated Alkali Vapor Cells

One of the most important steps in the fabrication of a micro-fabricated vapor cells is the encapsulation of the alkali metal in the interior of the vapor cell [56]. The alkali metal atoms tend to react with oxygen in the air, rendering the atomic sensor useless, because the bonding process between glass–silicon–glass usually requires a high temperature, while the alkali metals have a low melting point and are volatile. We grouped commonly utilized methods into the following five categories:

#### 2.3.1. Direct Filling Method of Alkali Metal Elements in the Physical Method

The packaging of atomic vapor cells is achieved by dropping an alkali metal into a vapor cell under vacuum conditions by a micro-pipette (Figure 12) [45,57,58,59,60].

In 2004, Liew et al. of NIST utilized the direct filling method of alkali metal to encapsulate the atomic vapor cell [61]. To avoid the reaction of alkali metal on contact with air, liquid Rb elements were directly dropped into the silicon holes and flushed with buffer gas.

In 2010, Fang et al. of Beihang University invented a process for the preparation of micro-fabricated atomic vapor cells [62], which enabled the on-chip integration of heating filaments, RF coils, and thermally isolated structures; the leakage rate of the vapor cell was less than 3 × 10^−8^ Pa·m^3^·s^−1^.

In 2011, Hasegawa et al. proposed a simple Cs filling method that involves locally heating a Cs dispenser with a high-power laser source after the sealing of cells [16]. The dispenser consisting of the cesium chromate and Zr-Al alloy mixture as well as the Cs is generated by laser activation. The mixture in this alkali dispenser has high chemical stability at temperatures up to 500 °C, and alkali vapors are generated at temperatures of about 700 °C. Changes in the buffer gas (in most cases, N_2_, Ne or Ar), pressure will directly affect the sensitivity of the vapor cell. Ne or Ar is employed because neither of these are affected by the Cs dispenser, unlike N_2_ which will be absorbed by the dispenser. This method was first utilized in atomic clocks, but since it requires the utilization of lasers to generate high-temperature heating of alkali metal dispensers to produce alkali metal, this would increase the complexity of the vapor cell fabrication. The laser power and irradiation time need to be precisely controlled. Not enough power or too short a time will make the whole process fail due to the reaction of alkali atoms with residues. Too much power or too long a time will produce a large amount of alkali metals, which will condense on the clear surface and affect the performance of the vapor cell. Therefore, this method has not been widely utilized in the application of atomic gyroscopes and atomic magnetometers.

In 2014, Ji et al. of Southeast University prepared a wafer-level spherical rubidium vapor cell by utilizing a combination of high-temperature thermoforming and a releasing agent [63], which first etched a hole array on a silicon wafer by utilizing dry etching and was then filled with an initiator for the first anodic bonding to produce a spherical vapor cell; then, a cavity array was etched by utilizing dry etching again, and it was finally filled with alkali metal atoms and buffer gas under vacuum and the second anodic bonding was performed to achieve a sealed vapor cell. The size of the alkali vapor cell reached 3 × 3 × 2 mm^3^.

The process of preparing atomic vapor cells through the direct filling of alkali metal avoids the introduction of impurities in the alkali vapor cells and enhances the performance of vapor cells with smaller planar dimensions. This method is suitable for the manufacture of gyroscopes, magnetometers, and atomic clocks because it is simple and easy to implement without introducing other impurities.

A common drawback of these methods is the micro-fluidic manipulation of liquid Cs and Rb. Because alkali metals are highly volatile, the micro-fluidic manipulation of Cs and Rb is not only a great technical challenge, but also limits the implementation of parallel processes, that is, only stepwise in a specific order, and the manipulation method may introduce non-alkali metal residues that affect the chemical stability of the cell. These drawbacks lead to poor device uniformity and, consequently, increased commercial costs.

#### 2.3.2. Alkali Metal Wax Bag Filling Method in the Physical Method

In 2005, Radhakrishnan et al. of Cornell University utilized the paraffin-wrapped Rb method to place alkali metal into an atomic vapor cell with a volume of 3–6 mm^3^, as shown in Figure 13 [57], where a SiN film was first deposited on the surface of the silicon pore, and then a wax package wrapped with alkali metal was adhered to the SiN film. After the alkali vapor cell was encapsulated, the alkali metal was evaporated into the silicon hole through the laser ablation of the silicon nitride and decomposition of the wax package. This packaging method is suitable for atomic clocks and atomic magnetometers.

In 2013, Chen et al. from Tsinghua University utilized paraffin-wrapped alkali metal to inject alkali metal into the atomic vapor cell [64], released rubidium or cesium into the cell by a high-power laser, and then sealed the atomic vapor cell by two-step low-temperature anodic bonding (<140 °C) and slowed down the collision of atoms with the cell wall by filling it with buffer gas and plating a uniform paraffin layer. The size of the alkali vapor cell was 6.5 × 4.5 × 2 mm^3^, with a leakage rate lower than 2.8 × 10^−7^ Pa·m^3^·s^−1^ to avoid the oxidation reaction of alkali metal in the anodic bonding process. At the same time, after the vapor cell was sealed, the alkali metal-wrapped coating was attached to the wall inside the cell, and the coating reduced the collision of alkali metal atoms with the cell wall. However, the melting times of silicon nitride and paraffin wax are not easy to control; the process is more complicated and cannot be performed in batches.

This method is simple and easy to carry out, and can realize the function of a vapor cell coating while introducing alkali metal, thus reducing the wall collision relaxation rate of alkali metal vapor. Although this method has many advantages, paraffin coatings cannot withstand high temperatures, and at about 90 degrees Celsius they will lose their function, and so it is not suitable for atomic gyroscopes that require high-temperature environments.

#### 2.3.3. Chemical Reaction to Form Alkali Metal 

Atomic vapor cells are made by packaging the compound mixture and by heating the atomic vapor cell to manufacture the desired alkali metal elements (Figure 14) [65,66].

In 2006, Knapp et al. of NIST utilized the reaction of BaN_6_ and RbCl in a glass ampoule to produce Rb alkali metal [67], which was finally sealed by anodic bonding with a vapor cell volume of 1 mm^3^. Their work allowed an atomic clock frequency stability of 6 × 10^−12^ at 1000 s and a long-term drift of less than 5 × 10^−11^ in one day. Their research focuses on the application of magnetometers. The chemical reaction equation is as follows:(1)$${\mathrm{BaN}}_{6}+\mathrm{RbCl}\to \mathrm{BaCl}+3N_{2}↑$$

This method first utilizes KOH wet etching to make a silicon hole in the silicon wafer, and then bonds the silicon and glass wafers together by an anodic bonding process, after which liquid Rb elements are dropped directly into the silicon hole in a low-temperature vacuum anaerobic environment that is then filled with buffer gas; finally, the atomic vapor cell is sealed by secondary anodic bonding. This method introduces impurities, and other designs are usually made to reduce or eliminate these effects.

In 2015, Ermak et al. of St. Petersburg Polytechnic University, Russia, utilized micro-fabrication techniques to simultaneously fabricate 97 atomic vapor cells [68], each with a size of 1.2 mm^3^, and test studies showed that the resonant signal had a spectral linewidth of up to 2–3 kHz and a signal-to-noise ratio of 1500 in a bandwidth of 1 Hz, allowing the atomic clock to achieve a frequency stability of 1 × 10^−11^ at 100 s. This article mainly focuses on the application of atomic clocks and does not describe the performance of these techniques in other applications.

The aforementioned chemical reaction fabricates alkali metal elements to prepare the atomic vapor cell without the direct manipulation of alkali metals during operation, which is simpler and avoids the problem of alkali metals being prone to chemical reactions with the outside air during the transfer process. However, this method may introduce residues of non-alkali metals and affect the light transmission rate. Therefore, the design scheme of the double-cavity structure is generally selected, and the reaction cavity and the working cavity are separated. This structural method is utilized in gyroscopes, magnetometers, and atomic clocks.

#### 2.3.4. UV Decomposition Method

The photodecomposition method mainly takes advantage of the fact that RbN_3_ and CsN_3_ are relatively stable at room temperature; alkali metal azide is sealed inside the atomic vapor cell, the alkali metal atoms are produced by thermal decomposition under laser irradiation, and the buffer gas enters the silicon cavity through the micro-fluidic channel to form an atomic vapor cell (Figure 15) [15,69,70,71,72,73,74,75,76].

In 2004, Liew et al. of NIST prepared an atomic vapor cell utilizing UV irradiation to decompose CsN_3_ into Cs and nitrogen. The cell utilized vacuum thermal evaporation to deposit CsN_3_ into an unclosed vapor cell; after the atomic vapor cell was packaged, UV irradiation was utilized to decompose CsN_3_ into Cs and nitrogen with a minimum size of 2 × 2 × 1.75 mm^3^ for a single vapor cell. The resonance parameters of the vapor cell CPT signal were tested, and the CPT signal linewidth was up to 2.7 kHz. This packaging method is more effective; this article mainly talks about its application in atomic clocks.

In 2015, Li et al., from the China Aerospace Control Instrumentation Institute, encapsulated a vapor cell utilizing the anodic bonding process [77] and utilized the method of photoirradiation decomposition to decompose RbN_3_ in the vapor cell into Rb and N_2_ to obtain a Rb atomic vapor cell with N_2_ as the buffer gas. This method of introducing alkali metals can produce alkali metals and buffer gas at the same time, but it takes a long time.

In general, the preparation of atomic vapor cells by photolysis is simple and convenient, and manufactured atomic vapor cells contain alkali metal atoms and N_2_, which can maintain their chemical purity for a long time; the pressure of the buffer gas can be strictly controlled by changing the parameters of photolysis. This method is suitable for gyroscopes, magnetometers, and atomic clocks. However, this method requires dozens of hours for the photolysis of RbN_3_ and CsN_3_.

#### 2.3.5. Electrochemical Decomposition

The basic idea of electrochemical decomposition is to place a special glass containing more alkali metals inside the encapsulated vapor cell, utilizing the external glass surface of the vapor cell to provide a source of Na ions to improve the electrical conductivity of the glass, utilize silicon as the negative electrode and glass as the positive electrode, and then finally Rb is released from the glass by electrolysis at high temperature (Figure 16).

In 2006, Gong et al. from Princeton University, USA, prepared atomic vapor cells utilizing the electrolytic generation of elements method [78]. In 2016, Ban et al. from the Department of Microengineering, Kyoto University, Japan, utilized an alkali metal source tablet (AMST) as an alkali metal distributor to fabricate alkali metal vapor cells for optically pumped magnetometer (OPM) arrays [68]. The basic principle is as follows: firstly, a mixture consisting of carbonate from cesium and oxide from boron was melted at 900 °C for 30 min to obtain a cesium-rich glass; secondly, a 2.5 mm diameter through-hole was drilled into a thick silicon wafer, with a thickness of 2.5 mm, by deep reactive ion etching, and then a 3 mm-thick Pyrex shallow slot was drilled into a 3 mm-thick Pyrex glass sheet to hold the cesium metal, and the vapor cell was sealed by an anodic bonding process with two identical glass sheets; finally, the anodic was connected to copper and the cathode was connected to silicon, and Cs monomers were produced by electrolysis. After the vapor cell was cooled, the cesium metal gathered in the vapor cell, and the CPT resonance linewidth of the atomic vapor cell was measured at 110 °C up to 12.3 kHz. Regarding the application of this alkali metal introduction method, there is no more scientific research to reference.

The electrochemical decomposition method to make atomic vapor cells can effectively control the filling amount of alkali metal and enable wafer-level batch fabrication, which greatly reduces the reduction of Na ions to Na atoms during anodic bonding and enhances the strength of anodic bonding. This method is suitable for gyroscopes, magnetometers, and atomic clocks. However, this method is more tedious to carry out, and different currents will produce different results and high costs; therefore, it has not been promoted and applied in recent years.

In Table 1, a summary of the different methods for packaging the alkali metal described is presented, highlighting the main advantages and disadvantages. 

## 3. Applications and Outlooks of Alkali Vapor Cells

### 3.1. Applications and Recent Progress on Alkali Vapor Cells

Micro-fabricated alkali vapor cells play an important role as the core components in various atomic sensors. The size of the cell directly determines the final volume of the atomic sensor, and the fabrication quality of the micro-fabricated alkali vapor cell also directly decides the final performance of the atomic sensor. The most common applications include atomic gyroscopes, atomic clocks, atomic magnetometers, etc.

#### 3.1.1. Application of Atomic Gyroscope to Micro-Fabricated Alkali Vapor Cells

In 2008, a chip-scale alkali vapor cell with an inverted pyramidal structure of the inner cavity was developed at NIST [45]. The atomic vapor cell is a three-layer-bonded glass–silicon–glass structure with a multilayer dielectric coating on the inner wall of the cell for beam reflection. This structure facilitates the integrated design of the instrument because the pump source and the detector are located in the same plane. The aforementioned atomic vapor cell was utilized in the development of a miniature NMR gyroscope by NIST in collaboration with the University of California, Irvine, USA, and the volume of the integrated core component was approximately 2 cm^3^ [44]. However, the study also showed that the atomic vapor cell prepared by this micro-machining process has many problems, such as difficulties in plating the inner wall reflection film layer and increasing the magnetic field gradient owing to the integrated heating coil, which needs to be further designed and improved. This method needs to be carried out in a high-temperature environment, and the thickness and uniformity of the spherical vapor cells produced are difficult to guarantee.

In 2007, a miniature atomic spherical vapor cell with a diameter of 700 μm was prepared by utilizing the blowing technique at the University of California, Irvine, USA [79]. The spherical alkali vapor cell prepared by this method reduced the effect of the self-produced magnetic field of the working medium on the spin-magnetic moment feeding of the nucleus, while providing a sufficient number of optical windows to avoid atomic access to the dead space. Moreover, compared to the design of the multi-layer-bonded structure, the spherical vapor cell does not need to be plated with an inner-wall reflective film layer, which reduces the process complexity and makes it easier to prepare in bulk. However, experimental studies have shown that, when these micro-fabricated spherical vapor cells are utilized in miniature NMR gyroscopes, the beam propagation direction changes several times during passage through the spherical cell, which is prone to stray light and polarization changes; this adversely affects the measurement signal.

Therefore, most atomic vapor cells with two-dimensional directional luminescence have a glass-cubic structure. The atomic polarization rate and nuclear spin relaxation time were enhanced by optimizing the gas composition ratio and improving the vapor cell cleanliness, flatness, and optical transparency rate. In published reports, the nuclear spin relaxation times of two isotopes, ^129^Xe and ^131^Xe, were greater than 22 s in an atomic vapor cell with a 1 × 1 × 1 mm^3^ inner cavity, developed by Northrop Grumman, USA, and 26 s in a 2 × 2 × 2 mm^3^ atomic vapor cell [80,81].

In 2017, Noor et al., of the Microsystems Laboratory, University of California, Irvine, USA, fabricated spherical micro-fabricated atomic vapor cells packaging alkali metals and inert gases by utilizing a glass-blowing process (Figure 17) [11]. Micro-heaters were designed to keep the alkali metal in the vapor state while minimizing the residual magnetic field, and an origami-type silicon structure with integrated optical reflectors made it possible to obtain an NMR gyroscope with an angular random wander of 0.1°/h and NMR magnetometer with a sensitivity of 10 fT/Hz^1/2^. This article describes the overall design of the NMR gyroscope and the design of the MEMS device components, which has a good reference value for the integration of the NMR gyroscope.

Micro-fabricated alkali metal vapor cells are essential for the creation of chip-level NMR gyroscopes and can achieve the goals of high precision, small size, and low cost, which lays a solid foundation for future applications of atomic gyroscopes, which have the potential for extensive development.

#### 3.1.2. Application of Atomic Magnetometer to Micro-Fabricated Alkali Vapor Cells

The low-temperature superconducting quantum interference device (SQUID) acts as an ultra-high-sensitivity magnetic field detector at 4 K, with a sensitivity of up to 1 fT/Hz^1/2^. It can be utilized to magnetically map the brain. Magnetometers based on the detection of Larmor feeds of atoms polarized by pumped light already approach similar sensitivity levels when utilizing large measurement volumes [82,83], but are much less sensitive in the more compact designs required for magnetic imaging applications [84]. A higher sensitivity and spatial resolution are possible with atomic magnetometers [1,3] for the non-invasive mapping of cortical modules. Article [23] describes a spin-exchange relaxation-free atomic magnetometer with a magnetic field sensitivity of 0.54 fT/Hz^1/2^ and a measurement volume of only 0.3 cm^3^. A theoretical analysis showed that the basic sensitivity limit of the device is below 0.01 fT/Hz^1/2^.

Ultra-sensitive magnetometers have a wide range of applications, including in condensed matter experiments [85], gravitational wave detection [86], nuclear magnetic resonance signal detection [87,88], paleomagnetic studies [89], non-destructive testing [90], and underwater detection. However, the most notable application of magnetic field sensors is in the field of biomagnetism [91,92], that is, the detection of weak magnetic fields produced by the human brain, heart, and other organs [28,29,93,94]. For example, the measuring of magnetic fields produced by the brain is conducted to diagnose epilepsy and study neural responses to auditory and visual stimuli. The SQUID sensors [95,96,97], which have so far dominated all of these applications, reach sensitivity levels of 0.9–1.4 fT/Hz^1/2^, with a pickup coil area of approximately 1 cm^2^. In the low-frequency range (<100 Hz) of biomagnetic studies, the noise is usually high, and the commercial SQUID magnetometer is typically approximately 5 fT/Hz^1/2^ [98], partly because of the magnetic noise produced by the conductive radiation shielding of the liquid helium Dewar bottle [99]. In addition to its high sensitivity, the SERF magnetometer described here does not require cryogenic cooling; this feature makes it very attractive for a wide range of applications, especially in environments outside the laboratory. The bandwidth and size of the magnetometer are well suited for detecting biofields. Magnetoencephalography (MEG) magnetometers require two technical improvements. First, the magnetometer must be placed into a larger magnetic shield (similar to the magnetic shield chambers currently utilized for MEG and SQUID magnetometers) [100] to accommodate patients within the shield. Second, more efficient heat dissipation and active cooling are also required to keep the outer surface of the sensor at room temperature.

The micro-fabricated magnetometer prepared by NIST in 2007 [101], as shown in Figure 18, utilizes a single-beam VCSEL scheme with a metric of 5 pT/Hz^1/2^. The micro-fabricated alkali vapor cell had a size of 1 × 2 × 1 mm^3^. This article proposes an integrated design scheme for magnetometers that has a high reference value in terms of the research of miniaturized magnetometers. The traditional sensor system utilized to measure the biological magnetic field is bulky, so this system may not suitable for the further development of magnetocardiography for clinical applications. In this study, the atomic magnetometer constructed by micro-manufacturing techniques is proposed, and the magnetocardiography of two mice is measured. 

In 2010, NIST prepared single-cavity and dual-cavity micro-fabricated vapor cells and the magnetometer test principle [102], as shown in Figure 19 (no prototype was demonstrated). Utilizing indium tin oxide (ITO) transparent heating film heating with a fiber-optic dual-beam scheme, they achieved a sensitivity of 10 fT/Hz^1/2^ in a 3 × 2 × 1 mm^3^ single-cavity experiment and a sensitivity of 10 fT/Hz^1/2^ in a 3 × 3 × 1 mm^3^ dual-cavity experiment. The sensitivity of the sensor with only Rb vapor was 5 fT/Hz^1/2^, and the sensitivity of the Rb/Ba mixture sensor was 8 fT/Hz^1/2^. In this study, the magnetometer has a single-beam scheme, and the design of a double-beam scheme requires structural changes.

Between 2012 and 2015, NIST updated and iterated a variety of micro-fabricated magnetometer prototypes [103,104], utilizing a modular design with a simple prototype structure, all with a single-beam model. Utilizing the SERF state to measure the magnetic field, changing from ITO electric heating [105] to an optically heated vapor cell scheme effectively improved the sensitivity index by an order of magnitude in the original scheme. An octagonal vacuum-encapsulated vapor cell was introduced, and the cell was supported and suspended by a cantilever beam, making the cell better insulated, while the vacuum insulation greatly reduced the possible high-temperature damage to the human body, simplifying the design scheme. ITO heating has the problem of high energy consumption, which goes against the development needs of low power consumption and miniaturization. 

In 2012, a single-beam fiber-introduction scheme was utilized, with a micro-fabricated alkali vapor cell size of 1.5 mm^3^ (Figure 20). This article reports the chip-scale atomic magnetometer measurement inductive and spontaneous magnetoencephalography and successfully detects magnetoencephalography signals on the scalp surface of a healthy human subject.

In 2015, a single-beam fiber-introduction scheme, where the insulation and sealed isolation of micro-fabricated alkali vapor cells, with a size of 1.5 mm^3^, were utilized to achieve a sensitivity of 30 fT/Hz^1/2^ by utilizing laser heating [104], was used to carry out a 25-channel magnetocardiography. A total of 25 sensor probes were mounted on the sensor strip for measurement, but only 16 sensor data points were utilized for analysis, and the noise level of the remaining sensors was too high for the data to be analyzed. Each pair of adjacent transducers were fixed parallel to each other and spaced 4 cm apart. The sensor fiber bundles were supported by an aluminum beam visible in the upper-left corner. These devices were in a magnetically shielded room, and the fiber-optic bundles were connected to the control device outside the room through an over-hole; thus, the size of the detection section could be reduced. This article made a magnetometer array for the study of magnetocardiography and successfully collected the cardiomagnetism signal.

In 2017, NIST implemented magnetic field imaging utilizing an OPM-MEMS magnetometer with a probe cross-section of less than 1 cm^2^ and a classical cantilever beam vapor cell structure [106], as shown in Figure 21, with a peripheral structure of 7.3 × 7.3 × 3.7 mm^3^ and an internal vapor cell of 1.5 × 1.5 × 1.5 mm^3^, by utilizing a laser-heated vapor cell technique, with a pumping light and heating light. Both the pumping and heating lasers were introduced by optical fibers (25 arrayed probes, using two distributed feedback laser (DFB) lasers and eight 1.5 μm lasers as the pumping and heating lasers, respectively). The micro-fabricated probes were light-pumped magnetometers with multiple modes, mainly in the zero-field mode. A total of 33 probes were assembled for the experiments, excluding the probes with a higher heating power and poor sensitivity, to conduct magnetic experiments on the brain for 21 probes with the best index. The full-field mode (Bell–Bloom magnetometer [107]) works in a geomagnetic environment and achieved a sensitivity test of 3 pT/Hz^1/2^ with a bandwidth of 1 kHz. This article has notable significance for the study of magnetometer arrays with the DBR laser-splitting scheme. A small-volume, arrayed fiber-coupled light-pumping atom magnetometer allows the sensor to have more flexibility in placement and density; this feature makes them very suitable in the imaging of various magnetic field sources, especially in applications in biomedicine.

In 2018, researchers at the University of Nottingham, UK, integrated a novel mobile magnetoencephalography device by utilizing a commercial SERF atomic magnetometer [108], as shown in Figure 22. The SERF atomic magnetometer was configured on the head of the subject for MEG testing by scanning the individual in 3D and customizing a 3D-printed helmet. The new mobile magnetoencephalography device is suitable for patients in various conditions and allows them to move freely while interacting with the real world. 

In 2011, Shang et al. of Southeast University proposed a spherical alkali vapor cell prepared by utilizing a chemical foaming process [109], which is prepared by first forming multiple cavities on the silicon wafer by etching, putting a foaming agent in the cavities, and then finally packaging them by covering the glass to the top of the foaming agent-filled silicon wafer cavities for anodic bonding. The average temperature is approximately 830–900 °C. The Pyrex glass softens at a high temperature and expands into a sphere under the action of the foaming agent, and finally, the whole body is cooled and annealed to form the cavity. This structure was tested morphologically, and it was found that the internal surface roughness of this spherical vapor cell was less than 1 nm, and the external surface roughness was 1.45 nm. This spherical vapor cell preparation method is low cost and provides a wafer-level vapor cell preparation method. 

In 2015, Ji et al. of Southeast University proposed a chip-scale magnetometer spherical glass hermetic structure, as shown in Figure 23 [110], where the upper part of the package is a wafer-level hemispherical glass cell array prepared by a controlled process combining thermoforming and chemical foaming, and the lower part is a glass wafer with conductive metal holes, with a package radius of 10 mm. The feasibility of the hermetic package was verified by measuring the average 0.2 °C internal temperature change with an external temperature change of 80–120 °C, which provided a new concept for the chip-scale atomic magnetometer package. The prepared package was also utilized to build a VCSEL laser frequency locking system, and the results locked the VCSEL frequency to the Rb atomic D_1_ line, illustrating the reliability of the package structure for application in chip-scale devices.

In 2017, the first integrated magnetic probe based on the SERF principle was studied by Southeast University, as shown in Figure 24 [111]. The volume of the sensor probe was 10 × 6.8 × 5 cm^3^, and the alkali vapor cell was a micro-fabricated vapor cell with a volume of 6 mm^3^. In this study, the fiber-optic introduction scheme was utilized; high-frequency electric heating and the effect of three different sets of buffer gas pressures on the sensitivity of the magnetometer were analyzed. It was concluded that the larger the buffer gas pressure, the higher the sensitivity; the higher the pumping optical power, the higher the sensitivity; and the smaller the vapor cell, the larger the impact of the wall collision, and the higher the required buffer gas pressure. This article mainly analyzes the influence of different buffer pressures on the performance of the magnetometer, but the number of samples is not enough.

In 2020, Ji et al. of Southeast University designed a pyramid-shaped micro-fabricated vapor cell [112], as shown in Figure 25. This structure changed the optical path from a vertical to parallel incidence compared to the conventional glass–silicon–glass structure, increasing the light range of the laser action inside the vapor cell. It was utilized for geomagnetic environment testing; the measured geomagnetic field magnitude was 47.85 μT, the noise power spectral density in the geomagnetic environment was 3.5 pT/Hz^1/2^ (1–2 Hz), and the slope of the linear region of the dispersion signal measured by the micro-fabricated vapor cell of this configuration was 15 times higher than that of the conventional glass–silicon–glass structure. This article mainly analyzes the influence of glass-blown vapor cells and flat vapor cells on the performance of magnetometers at the same volume, but it does not give a particularly detailed explanation for the reason behind this.

Currently, micro-fabricated alkali metal vapor cells enable the miniaturization of atomic magnetometers, which further makes high-resolution magnetoencephalography and magnetocardiography possible. The high sensitivity and small volume of micro-fabricated alkali metal vapor cells facilitated the development of atomic magnetometers with a small volume, high precision, and low cost, which highlights the great potential for magnetometers in applications for biological magnetic measurement and medical magnetic detection.

### 3.2. Conclusions and Outlooks of Alkali Vapor Cells

Driven by the demand of the sensing industry for the miniaturization, high precision, and integration of atomic sensors, the development of high-performance and miniaturized atomic vapor cells, which are the core components of atomic sensors, is gaining more and more attention. By utilizing various bonding techniques, high-performance alkali metal vapor cells can be achieved with excellent sealing, high transmission rates, and high surface flatness. At the same time, with the application of different alkali metal packaging methods, the encapsulation of alkali metals can be achieved efficiently at a low cost without introducing extra impurities or pollutants. The purpose of alkali metal packaging is to improve the purity and reduce the oxidation rate of the alkali metal, thus improving the performance of the final core components. In addition, as found by studies on coating technology for the inner wall of alkali metal vapor cells with various materials, the atomic wall collision relaxation rate can be reduced, thus improving the atomic polarization rate and relaxation time.

Based on the urgent demands of micro-fabricated alkali metal vapor cells for use in of atomic gyroscopes, atomic clocks, and atomic magnetometers, micro-fabricated alkali metal vapor cells are expected to be the most promising technologies that are drawing more and more attention. Research is being devoted to technical problems such as the gas ratios and precise control of the gas pressure in the micro-fabricated glass vapor cells in pursuit of efficient and low-cost fabrications of miniature alkali metal vapor cells for mass production.

## Figures and Tables

**Figure 1 biosensors-12-00165-f001:**
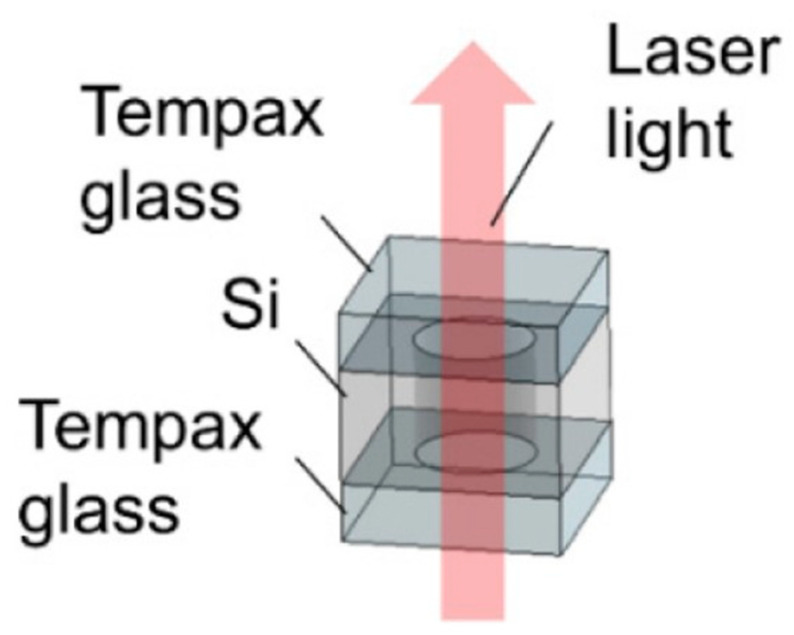
Silicon–glass anodically bonded alkali vapor cell. Reprinted from ref. [31].

**Figure 2 biosensors-12-00165-f002:**
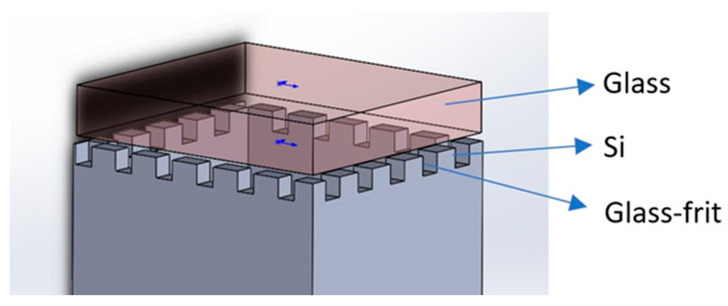
Sacrificial micro-channel-bonded alkali metal vapor cell.

**Figure 3 biosensors-12-00165-f003:**
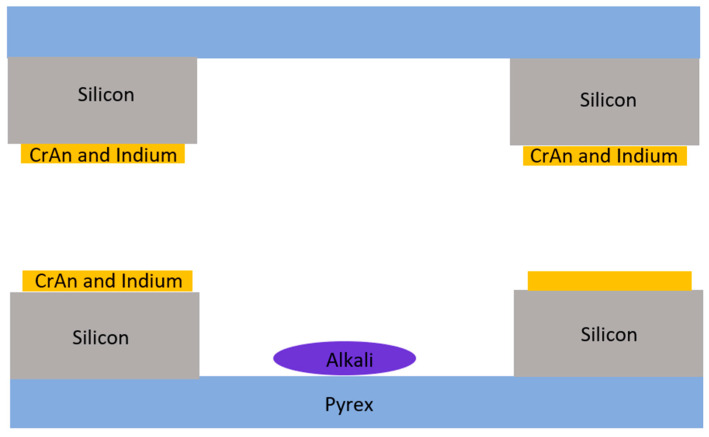
Thin-film-bonded alkali metal vapor cells.

**Figure 4 biosensors-12-00165-f004:**
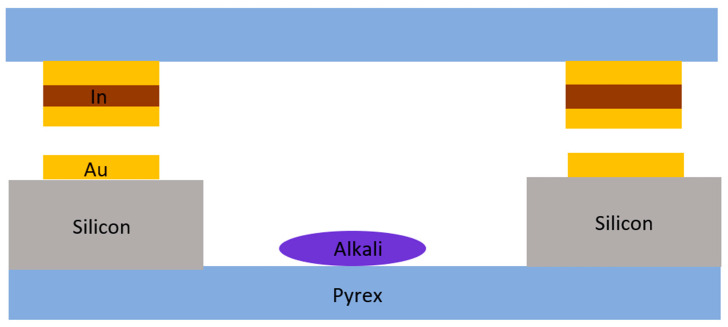
Thin-film-bonded alkali metal vapor cell with an asymmetric structure.

**Figure 5 biosensors-12-00165-f005:**
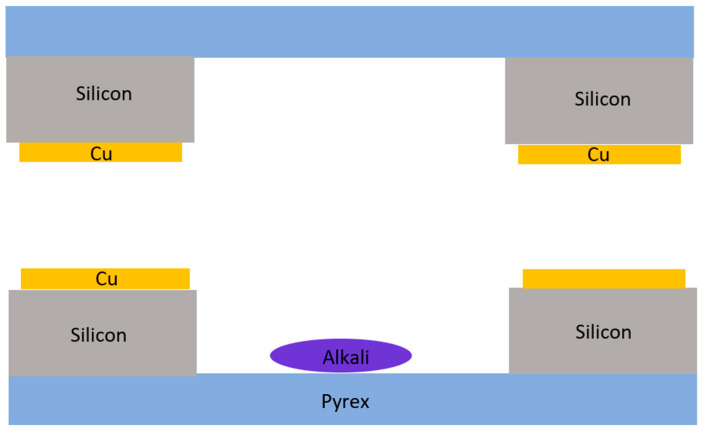
Cu-Cu-bonded alkali metal vapor cell.

**Figure 6 biosensors-12-00165-f006:**
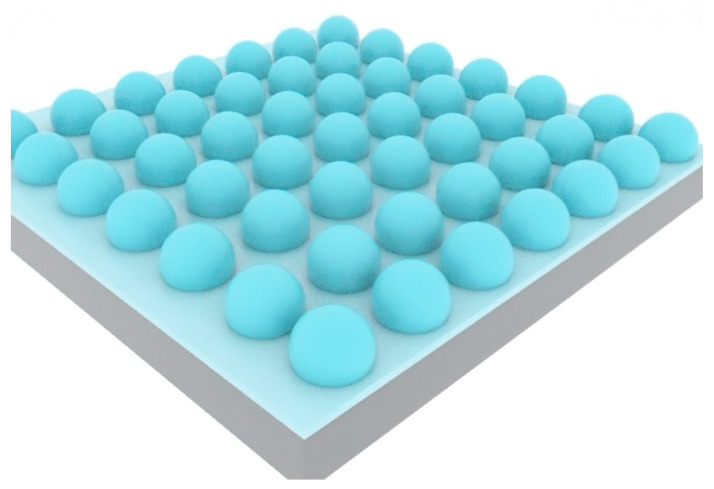
Wafer blowing of spherical alkali vapor cells.

**Figure 7 biosensors-12-00165-f007:**
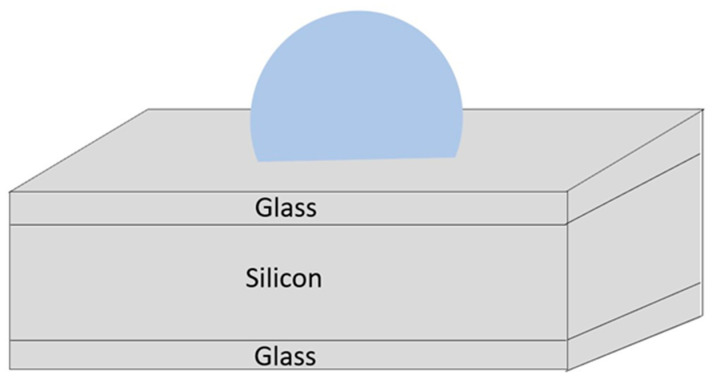
An individual spherical rubidium vapor cell with the external volume of 3 × 3 × 2 mm^3^.

**Figure 8 biosensors-12-00165-f008:**
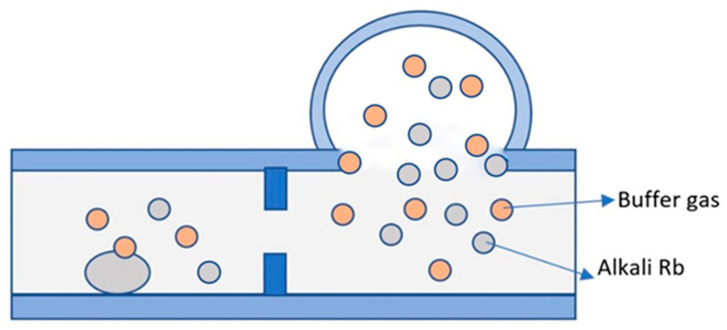
A schematic cross-sectional view of the micro-cells equipped with one spherical chamber and one planar chamber. The two cells are designed with the same volume of approximately 6 mm^3^.

**Figure 9 biosensors-12-00165-f009:**
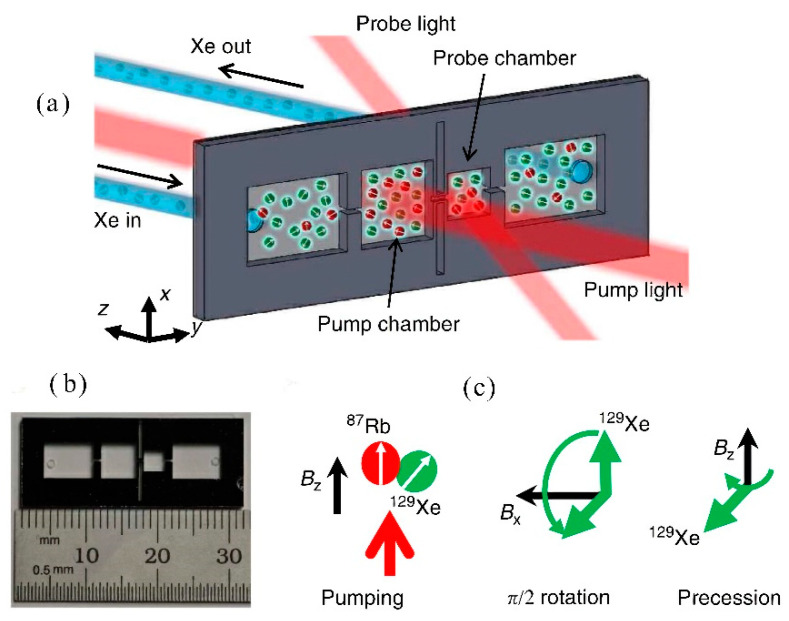
The micro-fluidic chip ^129^Xe. (**a**) A gas mixture containing 400 Torr N_2_ and 200 Torr Xe in natural isotopic abundance flows from a bulk gas manifold into the inlet chamber, through the pump and probe chambers, and out of the outlet chamber. The chip is loaded with 2 mg of ^87^Rb. The ^129^Xe then moves downstream, passes through a micro-channel into the probe chamber, and eventually exits the device through the output chamber. The optical characterization of the ^129^Xe polarization in the pump and probe chambers is carried out using the ensemble of ^87^Rb atoms in each chamber as in situ magnetometers. (**b**) The silicon chip footprint is 3 cm × 1 cm, with a thickness of 1 cm. The dimensions of the pump and probe chambers are 5 × 5 × 1 cm^3^ and 3 × 3 × 1 cm^3^, respectively, whereas that of the channel connecting the pump and probe chambers is 1 × 0.3 × 0.3 cm^3^. Two tall narrow grooves are etched from the middle of the chip to provide thermal isolation between the two sides of the device. (**c**) Pumping and probing sequence for ^129^Xe. Pumping is carried out continually in the pump chamber in the presence of a longitudinal field of B_z_ = 0.8 μT. Every 10–20 s, a transverse DC field of magnitude 5.3 μT is switched on for 4 ms to tip the ^129^Xe atoms onto the xy-plane and initiate the ^129^Xe precession about the longitudinal axis. Reprinted from ref. [53].

**Figure 10 biosensors-12-00165-f010:**
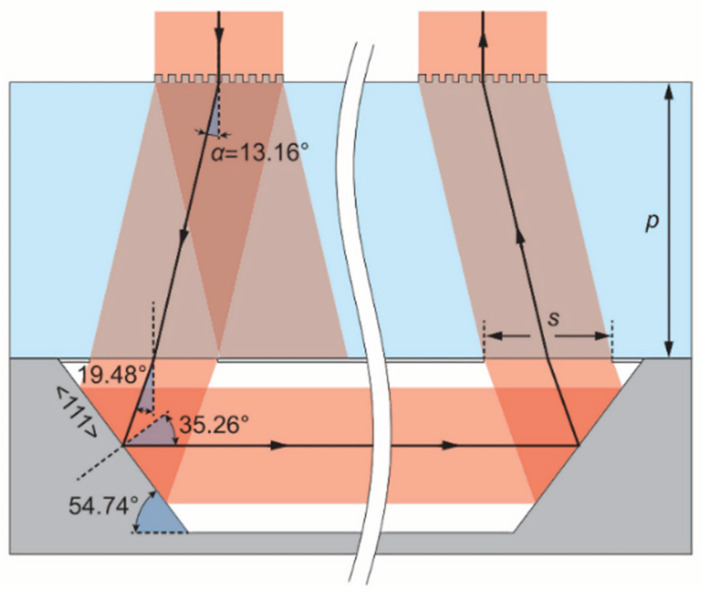
Routing of the beam in the cell. Reprinted from ref. [54].

**Figure 11 biosensors-12-00165-f011:**
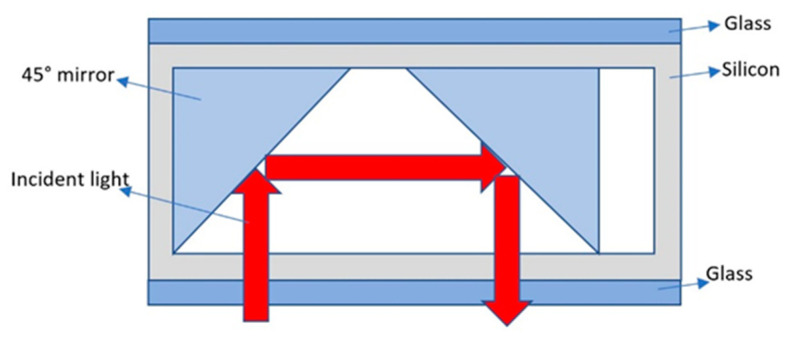
Concept and schematic view of the reflection-type vapor cell.

**Figure 12 biosensors-12-00165-f012:**
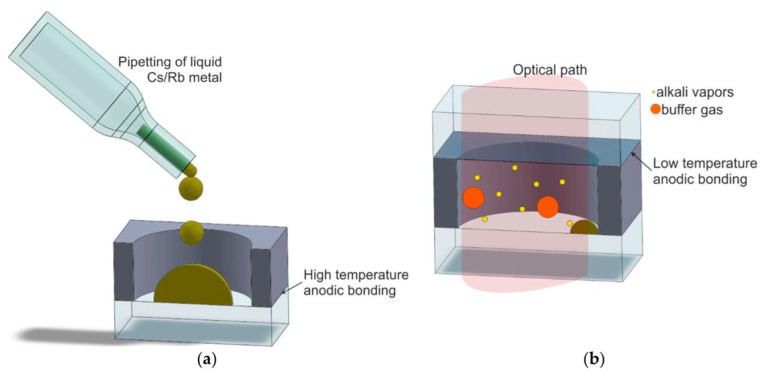
Atomic vapor cell preparation by direct filling of alkali metal monomers. (**a**) pipetting of liquid alkali metal, (**b**) optical path of alkali vapor cell. Reprinted from ref. [56].

**Figure 13 biosensors-12-00165-f013:**
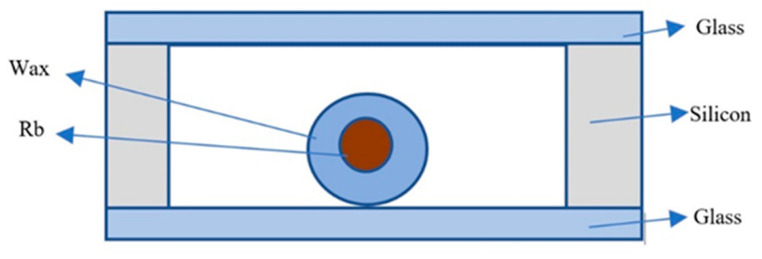
Flow chart of atomic vapor cell preparation by paraffin-wrapping method.

**Figure 14 biosensors-12-00165-f014:**
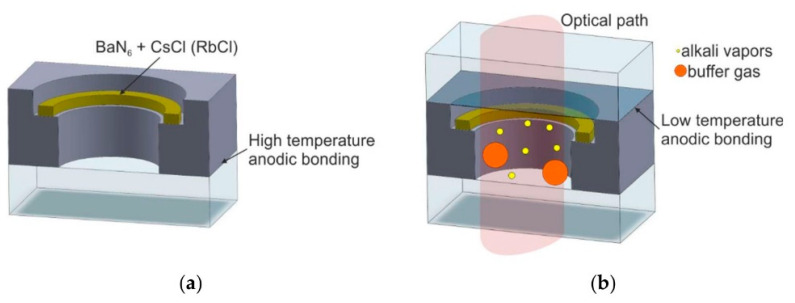
Vapor cell prepared by the chemical reaction to manufacture alkali monomers. (**a**) chemical reaction to form alkali metal, (**b**) optical path of alkali vapor cell. Reprinted from ref. [56].

**Figure 15 biosensors-12-00165-f015:**
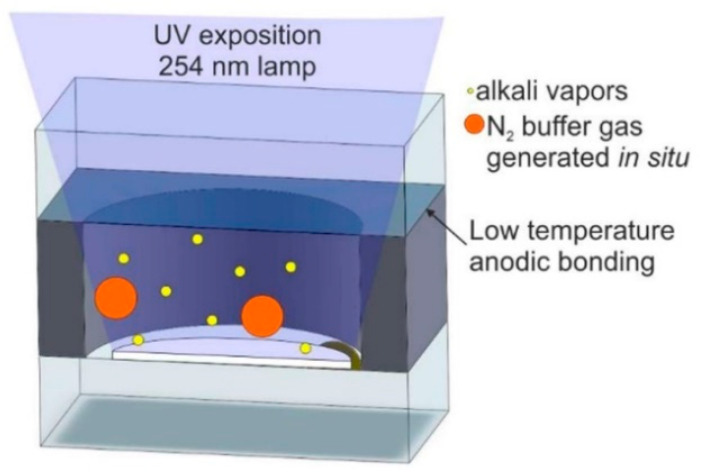
Process flow diagram for preparing atomic vapor cells using photodegradable CsN_3_ films. Reprinted from ref. [56].

**Figure 16 biosensors-12-00165-f016:**
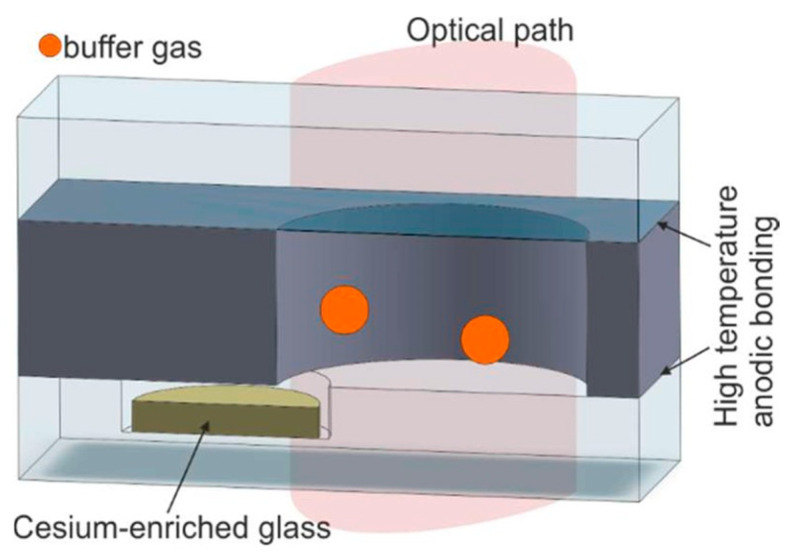
Process flow diagram of atomic vapor cell preparation by electrolysis. Reprinted from ref. [56].

**Figure 17 biosensors-12-00165-f017:**
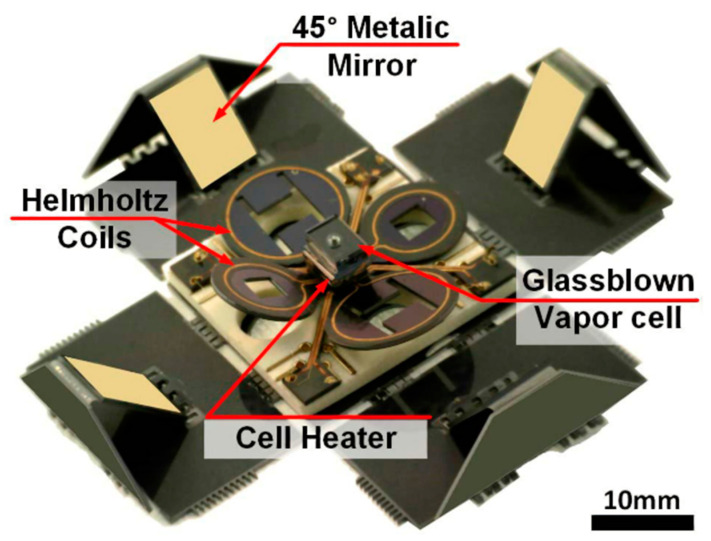
Partially folded NMR sensor prototype showing all components of the system. Reprinted from ref. [11].

**Figure 18 biosensors-12-00165-f018:**
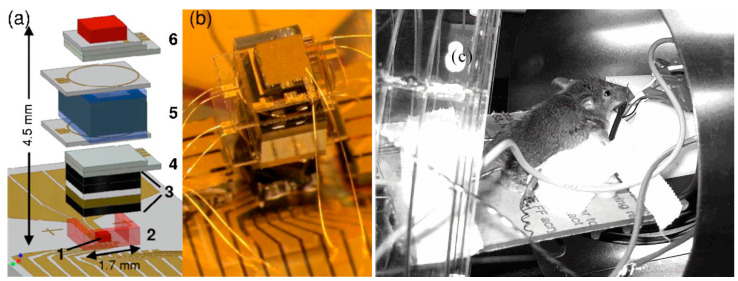
NIST micro-fabricated magnetometer scheme for measuring mouse heart magnetism (2007). (**a**,**b**) Magnetometer Structure, (**c**) Magnetometer applied to mouse magnetic measurement. Reprinted from ref. [101].

**Figure 19 biosensors-12-00165-f019:**
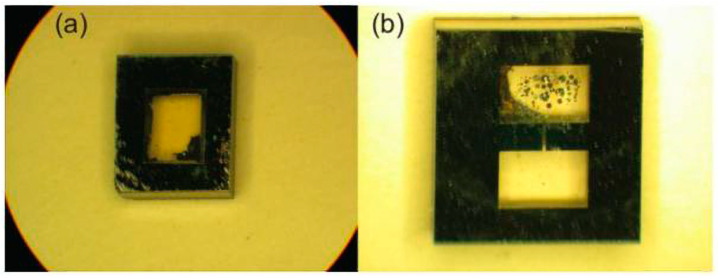
NIST dual-light magnetometer (2010). (**a**) A 3 × 2 × 1 mm^3^ single-chamber vapor cell; (**b**) double-chamber vapor cell: two chambers with a size of 3 × 2 × 1 mm^3^ connected by a 1 mm × 0.1 mm micro-channel. Reprinted from ref. [102].

**Figure 20 biosensors-12-00165-f020:**
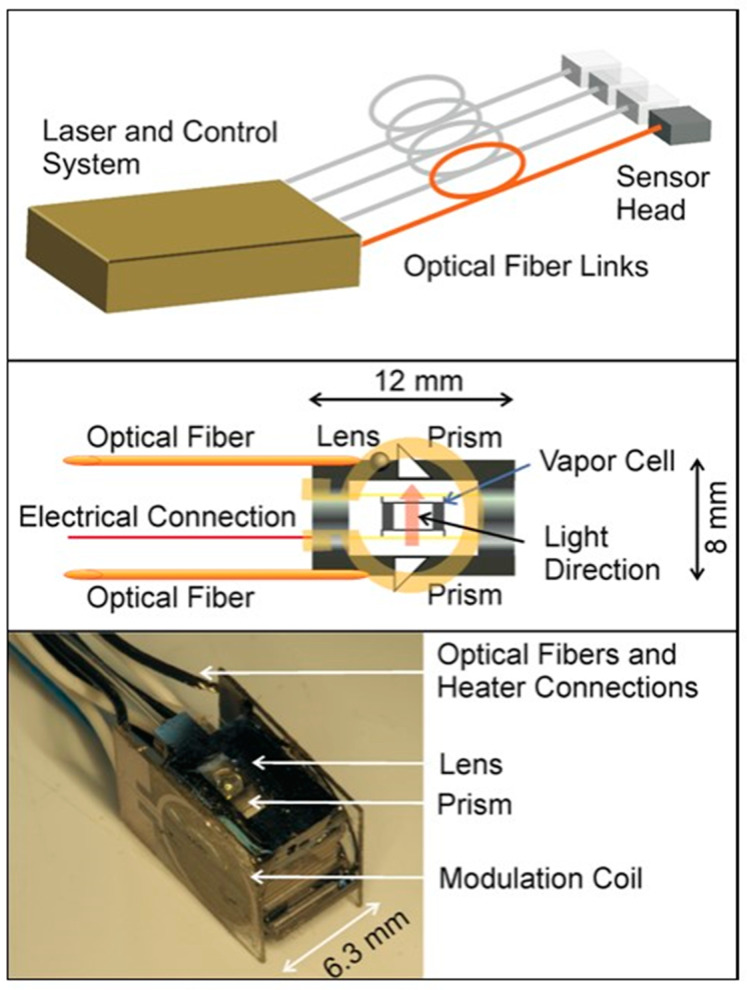
NIST micro-fabricated magnetometer (2012). Reprinted from ref. [103].

**Figure 21 biosensors-12-00165-f021:**
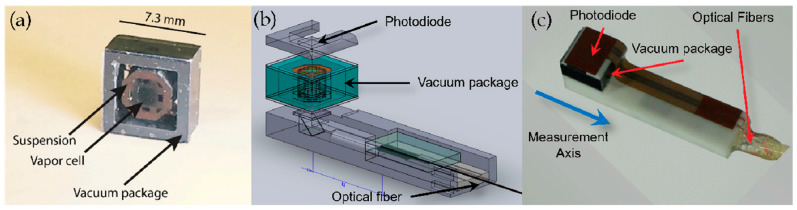
NIST OPM-micro-fabricated magnetometer for measuring brain magnetism (2017). (**a**) vapor cell, (**b**) Magnetometer schematic, (**c**) Magnetometer physical map. Reprinted from ref. [106].

**Figure 22 biosensors-12-00165-f022:**
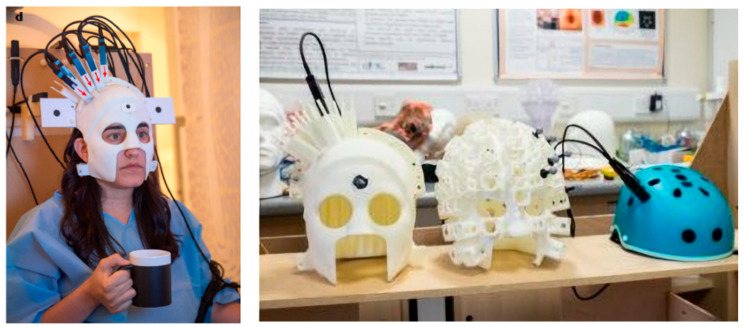
Magnetic brain probe array, Nottingham, UK (2018). Reprinted from ref. [108].

**Figure 23 biosensors-12-00165-f023:**
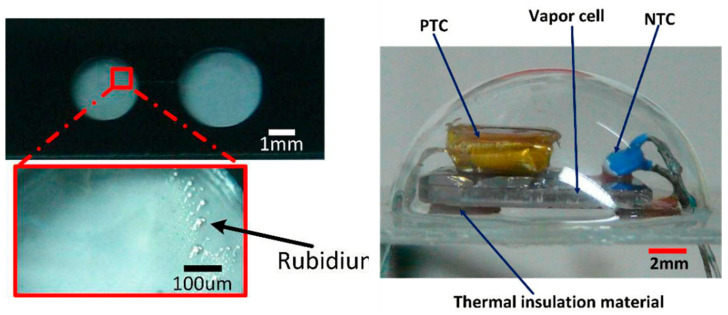
Glass encapsulation micro-fabricated vapor cell and heating and temperature measurement structure (Southeast University, 2015). Reprinted from ref. [110].

**Figure 24 biosensors-12-00165-f024:**
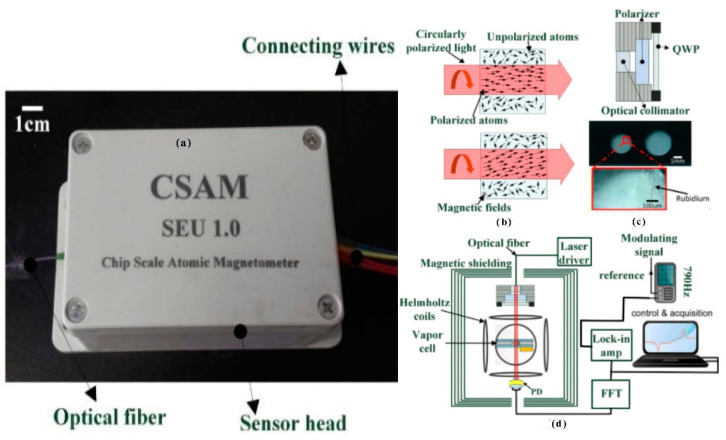
Magnetometer probe made utilizing micro-machining technology at Southeast University. (**a**) is atomic magnetometer, (**b**) is schematic, (**c**) is atom vapor cell, (**d**) is sensor structure diagram. Reprinted from ref. [111].

**Figure 25 biosensors-12-00165-f025:**
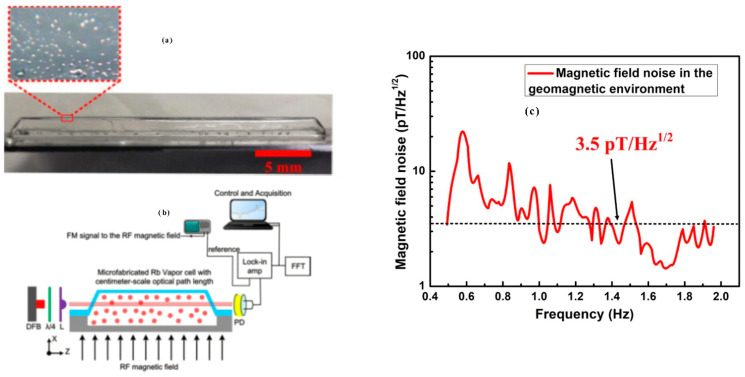
Pyramid-shaped micro-fabricated vapor cell (Southeastern University, 2020). (**a**) is the physical map of vapor cell, (**b**) is sensor structure diagram, (**c**) is the experimental data graph. Reprinted from ref. [112].

**Table 1 biosensors-12-00165-t001:** Comparison of the advantages and disadvantages of different encapsulation methods for alkali metals.

	Packaging Method	Advantages	Disadvantages
Physical methods	Direct filling of alkali metal monomers method	Avoids the introduction of impurities during the filling process and improves the performance of the alkali vapor cell	Extremely demanding in terms of equipment and environment, increasing the cost and complexity of the operation process
	Alkali metal wax package filling method	Oxidation during the filling process is avoided and the paraffin coating reduces the collision of alkali metal atoms with the inner wall of the vapor cell	The time control of the operating process of laser melting silicon nitride and paraffin wax is difficult, and the process is more complex; it cannot be batch-produced
Chemical methods	Chemical reaction to produce elements method	No need to operate directly on the alkali metal, avoiding the problem that the alkali metal is prone to chemical reaction with the outside world during the transfer process, and the operation is simpler	May introduce residues of non-alkali metals, affecting light transmission
	Ultraviolet photolysis method	Simple operation, no chemical impurities, can maintain the chemical purity of alkali metal for a long time, and can change the parameters to control the pressure of buffer gas	Longer time required for the photolysis of rubidium azide or cesium azide
	Electrochemical decomposition method	The amount of filling of base metal monomers can be well controlled, and wafer-level batch manufacturing can be achieved, enhancing the strength of anodic bonding	The operation is more tedious, and the process parameters have a large impact and high cost

## Data Availability

The data presented in this study are available on request from the corresponding author.

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
