# Peer review of "Recent Progress on Micro-Fabricated Alkali Metal Vapor Cells"

_biosensors, 2022, doi:10.3390/bios12030165_

Round 1

Reviewer 1 Report

see attached file

Author Response

Dear Editor and Reviewers,

Thanks for the comments concerning our manuscript entitled “Recent progress on micro-fabricated alkali metal vapor cells”. Those comments are all valuable and very helpful for improving our paper. We have studied these comments carefully and have made corrections. Modified parts in manuscript is highlighted. The corrections in the paper and the responds to the reviewer’s comments are all in the attachment.

Reviewer 2 Report

This resubmitted draft has been enriched by several figures that make the paper more readable and more pleasant. The content is good for publication, however some language and stylistic mistakes can be improved. I list in the following some the main issues:

p. 1

1) The sentence: “Nowadays, the sensing industry has become a major driving force for the miniaturization of atomic sensing devices, that in the near future, micro-fabricated alkali vapor cell technology may experience extensive developments.” is not clear. It seems there are several subjects, please correct.

 2) it is very strange to see references 1-3 and then to jump to 105 and 106 [1-3,105,106], please correct.

3) The authors do not mention cylindrical vapor cells, as far as I know they are very common especially in cm-scale vapor cell devices, so why not to mention them?

4) The authors make a wrong use of the term which throughout the paper. In general which refers to something which is immediately before:

The SQUID sensors [85–87], which have so far dominated all of these applications, …

This is a good way to use “which”. The following are bad use of the term which:

More importantly, traditional  alkali vapor cell fabricated through blowing method cannot be further miniaturized, which limits the further miniaturization of atomic sensors

At the same time, after the vapor  cell is sealed, the alkali metal wrapped coating is attached to the wall inside the cell, which  reduces the collision of alkali metal atoms with the cell wall

For example in the last sentence I understand that the cell reduces the collision of alkali metal atoms with the cell wall that of course it doesn’t make sense, instead the authors mean that the coating reduces the collision of alkali metal atoms with the cell wall, but the coating is far from which, so in this case which cannot be used. The are many cases in the whole paper please correct.

p.2

5) The sentence: “The application of micro-fabricated vapor cell is mainly in several fields” is a non sense, please eliminate mainly

6) Several major bonding methods exists to be corrected in Several major bonding methods exist

p. 3

7) please explain the meaning of NEG

8) the disadvantage is that the bonding environment involve high temperature, high voltage, under certain vacuum level…. To be corrected in the disadvantage is that the bonding environment involves high temperature, high voltage, under certain vacuum level….

9) Neuheitel to be corrected in Neuchatel, I suppose, please check

p. 4

10) I know that NMR stands for nuclear magnetic resonance but it should be defined

p. 5

11) The conventional planar structure of micro-fabricated alkali metal vapor cell has only  two planes for light transmission which only allows for propagation of single beam. While  for atomic sensors, dual-beam perpendicular propagation inside the cell is required as  circularly polarized light is needed for polarizing (pumping) alkali metal atoms, which  further hyperpolarizes noble gas atoms by mutual collisions between alkali metal atoms and noble gas atoms, and the other beam is incident perpendicular to the pumping light  through the vapor cell as a detection beam (which is usually linearly polarized) for the  detection of nucleon signal.

Bad paragraph with many which used improperly, the same for While, do not begin a sentence with while.

p. 6

12) Although there  have been a lot of researches on glass blowing vapor cells, square structure vapor cells are  mostly utilized in practical applications. Again, in my opinion cylindrical cells are mainly used, or do the authors refer to a specific applications?

p.7

13) clarify meaning  of SERF, it is said later in the paper but it should be moved before.

p. 12

14) say that CPT is for coherent population trapping

p. 13

15) However, this method requires a long  time for the photolysis of RbN3 and CsN3. Can the authors be more quantitative, what do they mean, hours, days, months…

16) meaning of OPM

p. 15

17) the following sentence is redundant, too many superconducting quantum interference…please simplify. The low-temperature superconducting quantum interference device (SQUID) superconducting quantum interferometer acts ….

18) It can be utilized to map the magnetoencephalography, probably the authors mean something like It can be utilized to map magnetically the brain, the map of the magnetoencephalography sounds a little bit obscure.

p. 17

19) meaning of ITO

p. 22

20) atomic vapor cells, which is the core…to be correct in atomic vapor cells, which are the core

Definitely I think that after addressing the previous minor points the paper can be published.

Author Response

Dear Editor and Reviewers,

Thanks for the comments concerning our manuscript entitled “Recent progress on micro-fabricated alkali metal vapor cells”. Those comments are all valuable and very helpful for improving our paper. We have studied these comments carefully and have made corrections. Modified parts in manuscript is highlighted. The corrections in the paper and the responds to the reviewer’s comments are all in the attachment.

This manuscript is a resubmission of an earlier submission. The following is a list of the peer review reports and author responses from that submission.

Round 1

Reviewer 1 Report

Summary

This paper summarizes the literature for microfabricated cells for atomic sensors such as atomic gyros, atomic magnetometers, and atomic clocks. It discusses the literature from few different aspects, the bonding methods, the light passing schemes and the alkali metal packaging methods. In terms of the bonding methods, the discussed methods are Glass-Silicon-Glass, sacrificial glass channels, and Metal film bonding. The discussed light passing schemes are glassblowing and square vapor cell with light routing to achieve the perpendicular beams. For the alkali packaging methods the paper compares the physical methods and the chemical methods and presents the pros and cons of each method.

The paper goes on and discusses some of the applications of the atomic cells and the outlook for alkali vapor cells.

Major Comments

  1. In the filling methods subsection, I believe adding the filling method using alkali dispensers should be considered and a proper citation to the first paper that introduced it on the micro fabricated cells need to be added. Also comparing this method and its suitability to different applications need to be mentioned.
  2. The paper mentions many of the recent progress of the micro-fabricated cells for atomic sensors, however it lacks the critical discussion of the sources that it cites.
  3. The paper lacks focus in terms of the application of the micro-fabricated cells. The author needs to perform an in depth analysis of the current literature based on the chosen scope of the paper. A good example for this kind of papers was published in 2019 and was focused on micro fabricated cells for atomic clocks, Knapkiewicz, Pawel. "Technological assessment of MEMS alkali vapor cells for atomic references." Micromachines1 (2019): 25.
  4. An example of the theme for in depth analysis is the discussion of the suitability of the cells for bio sensors which makes the paper suitable for this journal

Additional minor comments

  1. The author keeps on using of “Anodic bonding” and “anode bonding” interchangeably, it is common in the field to use anodic bonding so I advise the author to stick to that terminology to avoid any misunderstanding that might arise.
  2. The author should read about the bonding techniques and be able to differentiate between them (anodic bonding, thermocompression, glass frit bonding, eutectic bonding, etc.). Subsequently the manuscript needs to be revised based on that.
  3. The use of silica and silicon interchangeably is widely spread throughout the paper and this incorrect simply because silicon is the element which most wafers are made of and is used in these applications and silica is silicon oxide.
  4. Improper use of the phrase et al. The first author’s name should be mentioned and the et al. follows that. This can be seen in lines (188, 197, 205, and 215) and many more
  5. There are irrelevant references. Ref. 8-11 are not for atomic gyros.
  6. The references for the figures need to be checked. It looks like they are made manually and not updating with the addition and removal of references in the paper.
  7. Additional comments on the manuscript PDF.

Author Response

Dear Reviewer,

please see attachment.

Thank you.

Reviewer 2 Report

See attachment

Reviewer 3 Report

In this paper the authors provide a picture of the several technologies used to manufacture micro-fabricated alkali metal vapor cells. This kind of device is particularly important since micro-fabricated cells are used in many applications like gyroscopes, atomic clocks and magnetometers. Since this is a review, I don’t dispute the content as it is, rather I would give some suggestions in order to improve the presentation and readability of this work.

In the following some aspects of the paper that could be improved.

1) the main drawback of the paper is the lack of figures. I understand that since this is review and not an original work figures are of difficult access, but at least in some cases I think that it is possible to ask for the permission to publish figures taken from other journals. In general this permission is always given for a limited amount of figures. Therefore I suggest the authors to rekindle a little bit their work with some figures.

2) the English in in general acceptable but here and there can be improved. For example, line 11 “and etc”, I would remove and.

Line 22-23: Nowadays, the sensing industry has become major driving force for the….probably it is better to add “a”: Nowadays, the sensing industry has become a major driving force for the…

Line 111-112: While the glass softening point temperature is  high, which imposes high requirements for the experimental equipment and operation.  The sentence sounds incomplete.

Line 283-284:The atomic vapor cells are made by packaging the compound mixture, which is proportionally, in the atomic vapor cell by heating to manufacture the desired alkali metal  elements. The term proportionally does not sound properly in this sentence, please modify or correct.

3) line 29: since this is a review, it is nice to refer to other previous reviews, together with references [1-3] I would add:

Godone et al. High-performing vapor-cell frequency standards, Mar 2015 | RIVISTA DEL NUOVO CIMENTO 38 (3) , pp.133-171

Kitching Chip-scale atomic devices, Applied Physics reviews, vol. 5, 031302 (2018)

4) always line 29, I would say atomic gyroscopes, atomic clocks (plural)

5) matching the <111> sidewall, probably <111> is a typo

6) In 2006, Knapp et al. of NIST used BaN6 and RbCl to react in a glass ampoule to produce Rb alkali metal, followed by evaporation to drop Rb into an unsealed vapor cell [56], which was finally sealed by anodic bonding with a vapor cell volume of 1 mm3. The sentence as it is sounds a little bit confusing, can you try to simplify it?

7) line 406: the term SQUID should be defined here, not in line 425

8) line 431: please define SERF, I know that it may be obvious but this should be defined. Similarly, line 480: define DFB

9) the paper is focused on the cell manufacturing, rather than on its application. Only sometimes the authors describe what the sell is used for. This make the paper rather boring, In 2015…In 2017, In 2020..without describing too much about the application. Also, while the application on gyroscope and magnetometry are discussed, atomic clocks are neglected, it is not clear why. So the authors should consider to extend their work to take into account also clocks.

Definitely, I think that a paper resuming all the different techniques used worldwide to manufacture microcells can be useful to the scientific community, even if it is a little bit boring…

Therefore, I recommend the publication of this paper, provided the previous points are addressed in a new manuscript.

Author Response

Dear Reviewer,

Thank you.

Round 2

Reviewer 1 Report

Major Comments:

  1. In the filling methods subsection, I believe adding the filling method using alkali dispenser Pills like in [1] should be considered and a proper citation to the first paper that introduced it on the micro fabricated cells need to be added. Also comparing this method and its suitability to different applications need to be mentioned.
  2. It is not clear what more does this paper provides compared to a published work in 2019. "Pawel K. Technological assessment of MEMS alkali vapor cells for atomic references. Micromachines, 2019, 10(25): 1-20" 
    1. An added value could be discussing the literature on the microcells from the point of view of biosensors.

Minor Edits:

  1. The reference that compares the flat cells to the spherical cells came to incorrect conclusion based on their assumptions. the following needs to be added "The improvement in the flat cell over spherical cells is due to the fact that the optical path in the spherical cell is much longer and there is a greater interaction between the light and the gas in the spherical chamber."
  2. There is a figure that is repeated 3 times in the paper.

[1] Hasegawa, Madoka, et al. "Microfabrication of cesium vapor cells with buffer gas for MEMS atomic clocks." Sensors and Actuators A: Physical 167.2 (2011): 594-601.

Reviewer 2 Report

.